# Native adiponectin in serum binds to mammalian cells expressing T-cadherin, but not AdipoRs or calreticulin

**Shunbun Kita[1,2]\*, Shiro Fukuda[1], Norikazu Maeda[1,3], Iichiro Shimomura[1]**

[1]Department of Metabolic Medicine, Graduate School of Medicine, Osaka University, Osaka, Japan; [2]Department of Adipose Management, Graduate School of Medicine, Osaka University, Osaka, Japan; [3]Department of Metabolism and Atherosclerosis, Graduate School of Medicine, Osaka University, Osaka, Japan

**Abstract** Adiponectin is an adipocyte-derived atypically abundant circulating factor that protects various organs and tissues through its receptors, AdipoRs, calreticulin, and T-cadherin. To identify the major binding partner of circulating native adiponectin, we expressed these receptors on the surface of HEK293 cells. Adiponectin, either that in mouse or human serum, purified from serum, or produced by mammalian cells, bound to cells expressing T-cadherin, but not to those expressing AdipoR1 or calreticulin. The stable introduction of T-cadherin and AdipoR1 into CHO cells resulted in the cell surface localization of these receptors. Native adiponectin in serum bound to cells expressing T-cadherin, not to those expressing AdipoR1. The knockdown of T-cadherin, but not AdipoRs resulted in the significant attenuation of native adiponectin binding to C2C12 myotubes. Therefore, native adiponectin binding depended on the amount of T-cadherin expressed in HEK293 cells, CHO cells, and C2C12 myotubes. Collectively, our mammalian cell-based studies suggest that T-cadherin is the major binding partner of native adiponectin in serum.

DOI: https://doi.org/10.7554/eLife.48675.001

**\*For correspondence:**
shunkita@endmet.med.osaka-u.ac.jp

**Competing interests:** The authors declare that no competing interests exist.

## Introduction

Adiponectin is a circulating factor that is secreted from adipocytes (*Hu et al., 1996*; *Maeda et al., 1996*; *Nakano et al., 1996*; *Scherer et al., 1995*). Three types of receptors have been identified for this uniquely abundant circulating protein: AdipoRs, calreticulin, and T-cadherin.

AdipoR1 was discovered by expression cloning from a C2C12 myotube cDNA library through evaluations of the binding of biotin-labeled globular adiponectin produced in *E. coli* as bait (*Yamauchi et al., 2003*). It belongs to the PAQR receptor family, which has a 7-transmembrane domain with an opposite topology to the GPCR family (*Tang et al., 2005*).

Calreticulin is an endoplasmic reticulum (ER) luminal Ca2+-buffering chaperone that exists in the ER of cells (*Mendlov and Conconi, 2010*; *Michalak et al., 2009*). Cell surface exposure to calreticulin was initially reported to initiate the clearance of viable or apoptotic cells through binding to LRP on phagocytes (*Gardai et al., 2005*). A subsequent study demonstrated that adiponectin opsonized apoptotic cells, and the phagocytosis of cell corpses was mediated by the binding of adiponectin expressed in insect cells or *E. coli* to calreticulin on the macrophage cell surface (*Takemura et al., 2007*).

T-Cadherin was discovered by expression cloning from a C2C12 myotube cDNA library through evaluations of cell binding to coated recombinant adiponectin produced in HEK293 mammalian cells (*Hug et al., 2004*). It is classified as a member of the classical cadherins, such as E-cadherin and N-cadherin, due to its high homology of five extracellular cadherin repeats (*Hulpiau and van Roy, 2009*). However, T-cadherin is a unique cadherin with a glycosylphosphatidylinositol (GPI) anchor on

its C terminus and does not possess the transmembrane or intracellular domain generally required for signaling, which may hinder the function of T-cadherin as an adiponectin receptor.

The adiponectin protein accumulates in tissues, such as the heart, muscle, and vascular endothelium, through binding with T-cadherin (*Denzel et al., 2010*; *Fujishima et al., 2017*; *Matsuda et al., 2015*; *Parker-Duffen et al., 2013*; *Tanaka et al., 2019*). In T-cadherin null mice, the accumulation of the adiponectin protein was completely absent in these tissues, and, thus, HMW multimer adiponectin accumulated in blood (*Denzel et al., 2010*; *Matsuda et al., 2015*; *Parker-Duffen et al., 2013*). These findings were in contrast to the lack of significant changes in plasma adiponectin levels in AdipoR1- and R2-double knockout mice (*Yamauchi et al., 2007*). Human SNP studies including GWAS also indicated the importance of T-cadherin, but not AdipoRs or calreticulin, for plasma adiponectin levels, cardiovascular diseases, and glucose homeostasis (*Buniello et al., 2019*; *Chung et al., 2011*; *Dastani et al., 2012*; *Kitamoto et al., 2016*; *Morisaki et al., 2012*).

Numerous studies have attributed the functions of adiponectin to either of these receptors by showing a decrease in their functions via the genetic loss or mRNA knockdown of their receptors, including AdipoRs (*Straub and Scherer, 2019*; *Yamauchi et al., 2014*; *Yamauchi et al., 2003*; *Yamauchi et al., 2007*), calreticulin (*Takemura et al., 2007*), and T-cadherin (*Denzel et al., 2010*; *Fujishima et al., 2017*; *Parker-Duffen et al., 2013*; *Tanaka et al., 2019*). However, the direct binding of native adiponectin in biological fluids, such as serum, to its receptor warrants further study.

We herein demonstrated that native adiponectin in serum bound to cells expressing T-cadherin, but not to those expressing AdipoRs or calreticulin.

## Results and discussion

We investigated the binding of native adiponectin in serum to three adiponectin receptors by transiently overexpressing the cDNA of each receptor in HEK293 cells (*Figure 1A*). We directly examined mouse serum as the ligand solution, including the most native adiponectin, purified adiponectin from mouse serum (*Fukuda et al., 2017*), and full-length recombinant adiponectin produced in HEK293 cells. Native-PAGE showed differences in the distribution of molecular species between serum or purified adiponectin and recombinant adiponectin (*Figure 1B*). Recombinant adiponectin contained a lower amount of HMW multimer adiponectin than mouse serum and purified adiponectin from serum (*Figure 1B*). Transient transfection resulted in the successful overexpression of each receptor based on their expression levels quantified by RT-qPCR (*Figure 1C*). The treatment of cells with different preparations of adiponectin at 4°C for 1 hr resulted in the binding of prepared adiponectin only to cells expressing mouse T-cadherin (*Figure 1D*). Mouse serum and purified adiponectin showed similar binding, whereas recombinant adiponectin containing a lower amount of 6-mer and the HMW multimer exhibited markedly weaker binding (*Figure 1D*). The results of a native-PAGE analysis showed that more than 6-mer of multimeric adiponectin specifically bound to cells expressing mouse T-cadherin (*Figure 1—figure supplement 1*), which is consistent with previous findings (*Fukuda et al., 2017*; *Hug et al., 2004*). The dose-response study revealed the specific and saturable binding of native adiponectin in serum to cells expressing T-cadherin (*Figure 1E*).

Similar results were obtained when human cDNAs were overexpressed and the binding of adiponectin in human serum was assessed (*Figure 1—figure supplement 2A–D*). We previously reported that purified recombinant T-cadherin bound purified adiponectin with an affinity of KD = 1.0 nM (*Fukuda et al., 2017*). The present results showed the saturable binding of native adiponectin in serum to cells expressing T-cadherin, which is consistent with previous findings (*Fukuda et al., 2017*). Regarding AdipoR1 and calreticulin, three possibilities have been proposed: they were not effectively translated, were not effectively presented on the cell surface, or did not support the binding of native adiponectin in serum to cells.

To confirm that all receptors were effectively translated and presented on the cell surface, we expressed affinity-tagged receptors in HEK293 cells (*Figure 2A*). We added a high-affinity PA tag (*Fujii et al., 2014*) to the N termini of T-cadherin (*Ciatto et al., 2010*) and calreticulin (*Mendlov and Conconi, 2010*) and to the C terminus of AdipoR1 (*Yamauchi et al., 2014*) such that each receptor exposed the PA tag outside of the cell. Transiently expressing cells were surface-biotinylated, and lysates were applied to streptavidin beads. Total cell lysates (*Figure 2B* Total) and streptavidin-captured cell-surface proteins (*Figure 2B* Cell surface) were analyzed by Western blotting. The anti-PA-tag antibody NZ-1 detected similar levels of all receptors in total cell lysates and cell surface

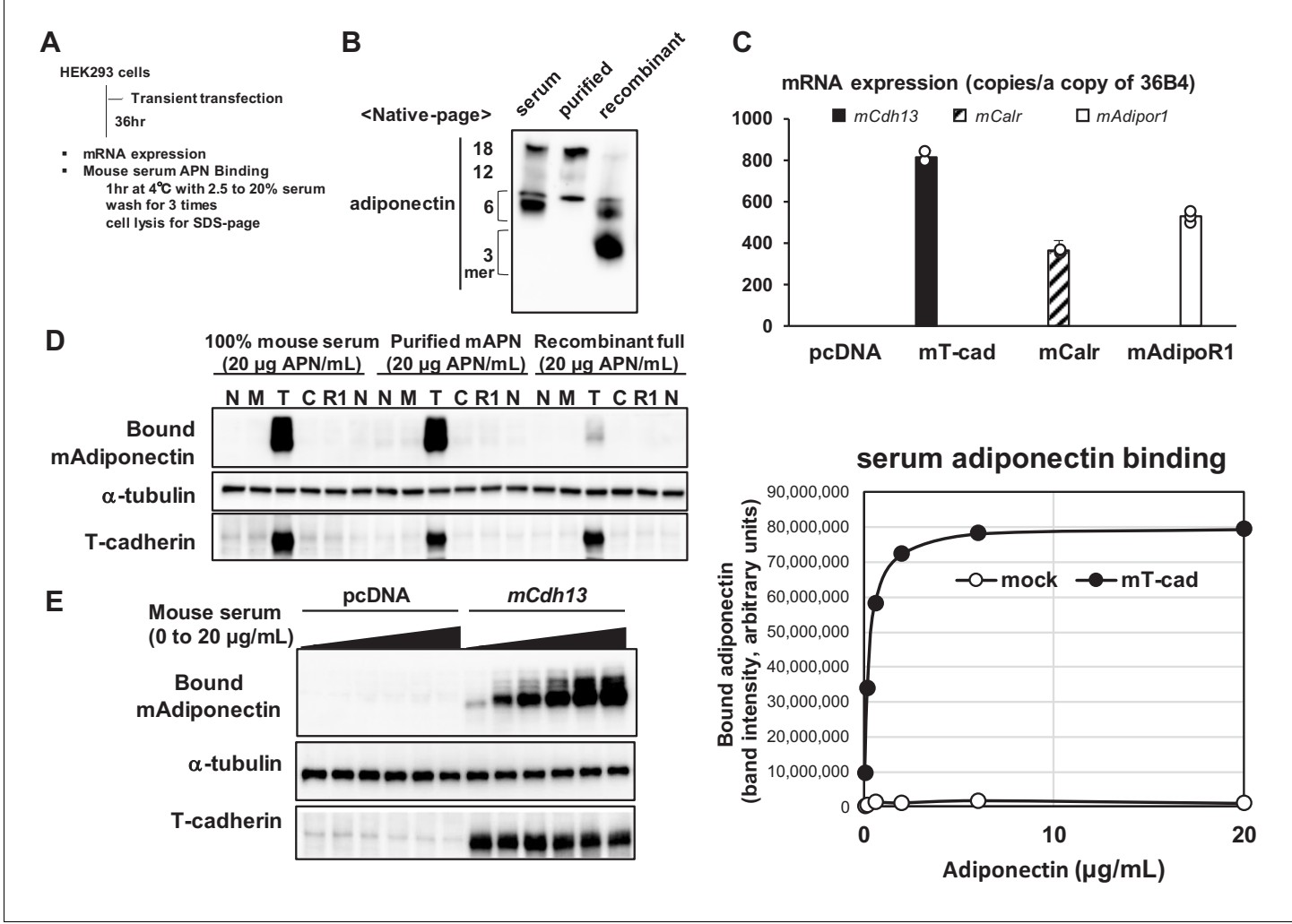

**Figure 1.** Mouse serum adiponectin binds only to the hek293 cells expressing t-cadherin. (A) Experimental outline. HEK293 cells were transfected with mammalian expression vectors coding mouse T-cadherin (*mCdh13*), Calreticulin (*mCalr*), or AdipoR1 (*mAdipor1*). (B) Native-page analysis of adiponectin preparations. Adiponectin concentrations were measured by ELISA and the equal amount (50 ng) of adiponectin was analyzed. (C) Absolute copy numbers of mRNA levels of mouse *mCdh13*, *mCalr*, *mAdipor1*, and *mAdipor2* were quantified. (D) Binding of adiponectin to HEK293 cells expressing none (N), mock (M), T-cadherin (T), or AdipoR1 (R1) (E) Dose-response cell-based binding study. Mouse serum (20 μg adiponectin/mL) was diluted and applied to the cells expressing mock or T-cadherin (left). The bound adiponectin was evaluated by blot intensity (right). Cell lysate following binding was separated by SDS-page and native-page. Essentially same results were obtained from more than three independent experiments.

DOI: https://doi.org/10.7554/eLife.48675.002

The following figure supplements are available for figure 1:

**Figure supplement 1.** Mouse serum adiponectin binding.
DOI: https://doi.org/10.7554/eLife.48675.003

**Figure supplement 2.** Human serum adiponectin binds only to the hek293 cells expressing t-cadherin.
DOI: https://doi.org/10.7554/eLife.48675.004

fractions, indicating that these receptors were successfully translated and expressed on the cell surface. Although AdipoR1 poorly migrated on the SDS-PAGE gel, this may have been due to heat-induced protein crosslinking or aggregate formation during sample processing (*Tanford and Reynolds, 1976*). The correct sorting of this protein to the cell surface suggested that AdipoR1 was expressed with the correct conformation on the cell surface. Under these conditions, a binding study with mouse serum revealed the dose-dependent binding of native adiponectin to cells expressing PA-tagged T-cadherin, but not to those expressing PA-tagged calreticulin or AdipoR1 (*Figure 2C*).

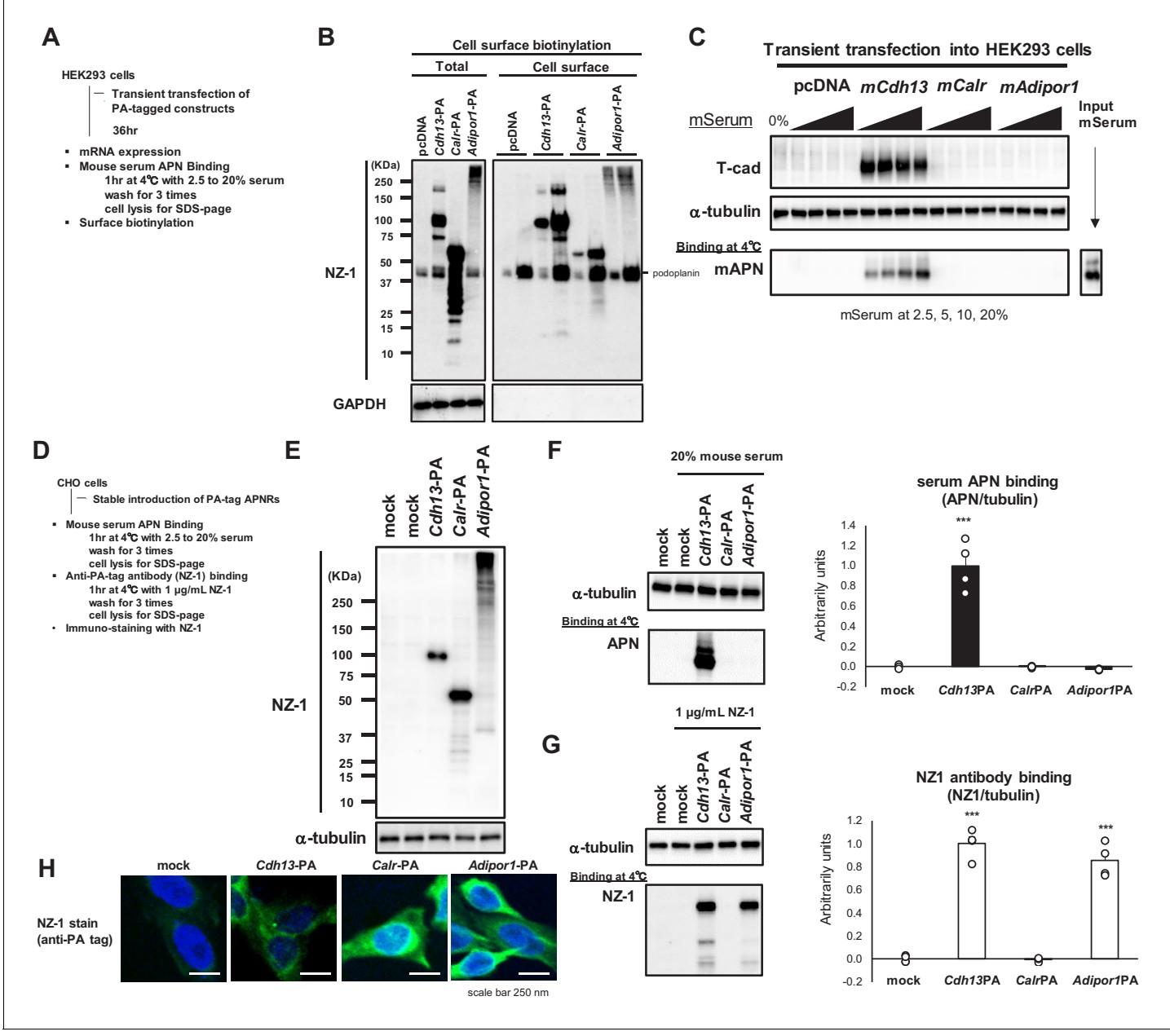

**Figure 2.** Cell surface expression of adiponectin receptors. (**A**) Experimental outline of transient expression in HEK293 cells. (**B**) Surface protein biotinylation analysis. Cell surface biotinylated proteins trapped on Streptavidin beads were eluted and analyzed in SDS-page in two lanes; x1 and x5 concentrations. Note that PA-tag antibody NZ-1 react with human podoplanin (40 KDa) in addition to PA-tagged proteins. GAPDH; control cytosolic protein. (**C**) Binding of mouse adiponectin in mouse serum (MS) to HEK293 cells. (**D**) Experimental outline of stable CHO cells expressing adiponectin receptors. (**E**) stable expressions detected by NZ-1. (**F**) Binding of mouse adiponectin in mouse serum to CHO cells. Adiponectin-binding from western blots (left) was calculated and expressed in right graph. Data are mean ± SEM. n = 4 ***p<0.01 (unpaired t-test). (**G**) Binding of NZ-1 antibody to CHO cells. NZ-1-binding from western blots (left) was calculated and expressed in right graph. Data are mean ± SEM. n = 4 ***p<0.001 (unpaired t-test). (**H**) Confocal immunofluorescence micrographs of CHO cells stained with anti-PA tag antibody NZ-1 (green). Cell nuclei were counterstained with DAPI (blue). Scale bars = 250 nm.

DOI: https://doi.org/10.7554/eLife.48675.005

Taken together with the results of the overexpression study (*Figure 1*), native adiponectin in serum bound to cells expressing T-cadherin, but not those expressing calreticulin or AdipoR1.

We then generated stably expressing CHO cells (*Figure 2D*). The successful expression of the receptors in these cells was confirmed by Western blotting (*Figure 2E*). Serum adiponectin binding

was only observed on cells expressing PA-tagged T-cadherin (*Figure 2F*). In contrast, the binding of the PA tag antibody NZ-1 to intact cells was detected on cells expressing PA-tagged T-cadherin and AdipoR1 (*Figure 2G*). These results demonstrated that AdipoR1 was stably expressed on the cell surface with the expected topology of the PA tag outside of cells, but did not induce the binding of native adiponectin in serum. Calreticulin was not recognized by NZ-1 in the binding study (*Figure 2G*), suggesting that the PA tag at the N terminus of calreticulin was not accessible by NZ-1.

This may have been due to some steric hindrance because calreticulin with the N-terminal PA tag was detected on Western blots (*Figure 2E*) and immunostaining of fixed and permeabilized cells by NZ-1 (*Figure 2H*). Calreticulin is essentially an ER-resident protein (*Gardai et al., 2005*; *Takemura et al., 2007*). Therefore, stably expressed calreticulin in CHO cells may not have been sorted to the cell surface.

Based on these different approaches, we concluded that the expression of AdipoR1 may not promote native adiponectin binding. We also concluded that if calreticulin is expressed on the cell surface, it may not promote native adiponectin binding. The present results demonstrated that only the expression of T-cadherin on the cell surface may increase the binding of native adiponectin.

Since the above studies employed artificial expression systems, and AdipoR1 and T-cadherin were both identified from the C2C12 cDNA library (*Hug et al., 2004*; *Yamauchi et al., 2003*), we examined native adiponectin binding to C2C12 myotubes (*Figure 3A*). The absolute expression level of T-cadherin mRNA in differentiated C2C12 myotubes was markedly higher than that of AdipoRs (*Figure 3B*). We investigated the knockdown effects of these receptors on the binding of serum-containing adiponectin in C2C12 myotubes (*Figure 3C,D*). The introduction of RNAi before differentiation resulted in the effective knockdown of T-cadherin, AdipoR1, or AdipoR1 and R2 after 3 days of differentiation (*Figure 3C*). The knockdown of T-cadherin resulted in significant reductions in adiponectin binding (*Figure 3D,E*). The knockdown of AdipoR1 or both AdipoR1 and R2 did not significantly reduce adiponectin binding (*Figure 3D,E*). Although slight decreases were observed in adiponectin binding by the knockdown of AdipoR1 or both AdipoR1 and R2, the strong correlation ($R^2 = 0.9896$) between T-cadherin expression and adiponectin binding at all experimental points suggested that these changes were due to the decreased expression of T-cadherin (*Figure 3F,G*). Collectively, these results indicated that native adiponectin binding also depends on the amount of T-cadherin expressed in C2C12 myotubes.

Here, we simultaneously compared three adiponectin receptors. The expression of T-cadherin gave adiponectin binding, which is consistent with our previous finding showing that recombinant T-cadherin binds HMW adiponectin in a 1: one ratio with high affinity (*Fukuda et al., 2017*). There was no detectable binding of adiponectin on the AdipoR or calreticulin. AdipoR was discovered by expression cloning. The overexpression of AdipoR was expected to promote ligand binding. The initial discovery of AdipoRs also indicated that HEK293 cells overexpressing AdipoRs bound *E. coli* recombinant globular adiponectin (*Yamauchi et al., 2003*). Therefore, difficulties are associated with speculating about the much weaker affinity or the requirement for some 'accessory' proteins to confer adiponectin binding activity to AdipoRs.

Since numerous studies indicated that AdipoRs mediate adiponectin signaling in a number of cell types, an additional activating mechanism for AdipoRs by adiponectin, that is, the reductive or proteolytic generation of the trimer, monomer, and/or globular adiponectin, may exist. The results of the present study, which focused on the direct binding of native HMW adiponectin, may indicate the activation of AdipoRs by low-molecular-weight (LMW) forms. On the other hand, in the case of calreticulin, this was evidenced by a neutralizing antibody treatment inhibiting the adiponectin interaction with cells (*Takemura et al., 2007*). Therefore, some 'accessory' proteins may be required to confer adiponectin binding activity to calreticulin.

T-cadherin binds clinically important HMW multimer adiponectin with high affinity (*Fukuda et al., 2017*) and mediates adiponectin-induced exosome biogenesis and ceramide efflux to exosomes (*Obata et al., 2018*). Such exosome-effect required T-cadherin, but not AdipoRs. The exosome mediates cell-cell communication by transferring signaling components such as microRNAs, bioactive lipids, and proteins in addition to its role in waste disposal (*Kita et al., 2019*; *van Niel et al., 2018*). The stimulation of exosome biogenesis by adiponectin was the first demonstration of a secreted factor modulating exosome biogenesis and secretion (*Obata et al., 2018*). Adiponectin in serum or purified native adiponectin together with T-cadherin accumulated inside multivesicular bodies, the site of exosome generation, both in cultured endothelial cells and the in vivo wild-type

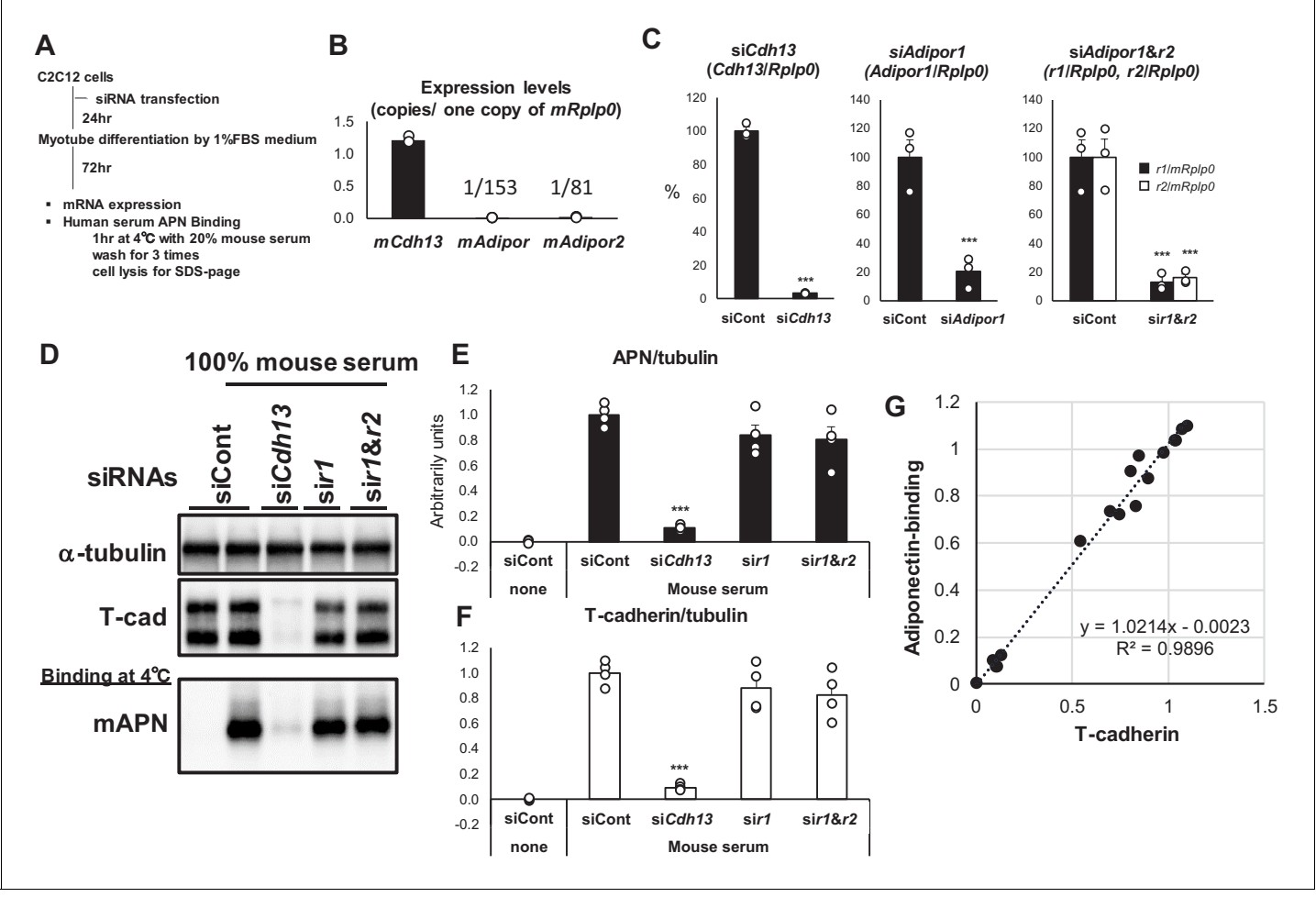

**Figure 3.** Knockdown of T-cadherin but not Adipors nor calreticulin affected native adiponectin binding to c2c12 myotubes. (**A**) Experimental outline. (**B**) Absolute expression levels of T-cadherin (*mCdh13*), AdipoR1 (*mAdipor1*), and AdipoR2 (*mAdipor2*) in differentiated C2C12 myotubes. (**C**) Knockdown efficiencies of adiponectin receptors. Data are mean ± SEM. n = 3 ***p<0.001 (unpaired t-test). (**D**) Binding of native adiponectin in mouse serum to C2C12 cells. Bound adiponectin was shown in representative western blot with a-tubulin as internal control and T-cadherin. (**E**) Adiponectin-binding. Data are mean ± SEM. n = 4 ***p<0.001 (unpaired t-test). (**F**) Amount of T-cadherin protein expression. Data are mean ± SEM. n = 4 ***p<0.001 (unpaired t-test). (**G**) Correlation between bound adiponectin and T-cadherin protein expression. n = 16 a linear regression r2 = 0.9896.
DOI: https://doi.org/10.7554/eLife.48675.006

mouse aorta (*Obata et al., 2018*). The systemic level of exosomes in blood was decreased by approximately 50% following the genetic loss of adiponectin or T-cadherin, but was increased by the overexpression of adiponectin in mice (*Obata et al., 2018*). The molecular mechanisms by which adiponectin stimulates exosome biogenesis are currently under investigation. We speculate that native HMW adiponectin with its multimeric structure may cause the higher-order clustering of T-cadherin, a membrane-anchored protein that resides in lipid rafts, and, thus, may stimulate exosome biogenesis. Adiponectin-induced increases in exosome biogenesis were not restricted to cultured endothelial cells because they were also observed in C2C12-differentiated myotubes (*Tanaka et al., 2019*) These findings support T-cadherin mediating adiponectin functions as a receptor for native HMW multimer adiponectin (*Kita et al., 2019*). The present results may further contribute to clarifying the activating mechanism of AdipoRs by LMW adiponectin, generated by reduction or proteolytic cleavage, followed by HMW adiponectin binding to cell surface T-cadherin.

# Materials and methods

## Key resources table

| Reagent type (species) or resource | Designation | Source or reference | Identifiers | Additional information |
|---|---|---|---|---|
| Antibody | anti-mouse adiponectin | R and D | AF1119 | goat polyclonal WB (1:5000) |
| Antibody | anti-human adiponectin | R and D | AF1065 | goat polyclonal WB (1:5000) |
| Antibody | anti-T-cadherin | R and D | AF3264 | goat polyclonal WB (1:5000) |
| Antibody | anti-α-tubulin | Cell Signaling | 11H10 | rabbit polyclonal WB (1:1000) |
| Antibody | anti-PA-tag (NZ-1) | FUJIFILM | 012–25863 | rat monoclonal WB (1:1000) |
| Commercial assay, kit | Cell Surface Biotinylation Kit | Thermo Fisher (Pierce) | 89881 | |
| Biological sample (*Mus musculus*) | Serum | CLEA Japan | C57BL6J jcl | Freshly isolated from C57BL6J mice, male |
| Biological sample (*Homo sapiens*) | Serum | | | Freshly isolated from healthy volunteers, male |
| Peptide, recombinant protein | Full-length mammalian recombinant mouse adiponectin | BioVendor | RD272023100 | |
| Peptide, recombinant protein | high-molecular weight purified mouse adiponectin | *Fukuda et al., 2017* | | |
| Sequence-based reagent | *mRplp0*_Fw | Gene Design | | GGCCAATAAGGTGCCAGCT |
| Sequence-based reagent | *mRplp0*_Rv | Gene Design | | TGATCAGCCCGAAGGAGAAG |
| Sequence-based reagent | *Adipor1*_Fw | Gene Design | | AATGGGGCTCCTTCTGGTAAC |
| Sequence-based reagent | *Adipor1*_Rv | Gene Design | | GGATGACTCTCCAACGTCCCT |
| Sequence-based reagent | *Adipor2*_Fw | Gene Design | | GGAGTGTTCGTGGGCTTAGG |
| Sequence-based reagent | *Adipor2*_Rv | Gene Design | | GCAGCTCCGGTGATATAGAGG |
| Sequence-based reagent | *mCdh13*_Fw | Gene Design | | GCCCTCGTGAGCCTTCTTC |
| Sequence-based reagent | *mCdh13*_Rv | Gene Design | | CACCCTGAGGTCCGTGATGT |
| Sequence-based reagent | *mCalr*_Fw | Gene Design | | AAGATGCCCGATTTTACGCAC |
| Sequence-based reagent | *mCalr*_Rv | Gene Design | | CCCACAGTCGATATTCTGCTC |
| Sequence-based reagent | *hRPLP0*_Fw | Gene Design | | GGCGACCTGGAAGTCCAACT |
| Sequence-based reagent | *hRPLP0*_Rv | Gene Design | | CCATCAGCACCACAGCCTTC |
| Sequence-based reagent | *hCDH13*_F | Gene Design | | AGTGTTCCATATCAATCAGCCAG |

*Continued on next page*

*Continued*

| Reagent type (species) or resource | Designation | Source or reference | Identifiers | Additional information |
|---|---|---|---|---|
| Sequence-based reagent | hCDH13_R | Gene Design | | CGAGACCTCATAGCGTAGCTT |
| Sequence-based reagent | hADIPOR1_F | Gene Design | | TCCTGCCAGTAACAGGGAAG |
| Sequence-based reagent | hADIPOR1_R | Gene Design | | GGTTGGCGATTACCCGTTTG |
| Sequence-based reagent | hADIPOR2_F | Gene Design | | CTGGATGGTACACGAAGAGGT |
| Sequence-based reagent | hADIPOR2_R | Gene Design | | TGGGCTTGTAAGAGAGGGGAC |
| Sequence-based reagent | hCALR_Fw | Gene Design | | CTCTGTCGGCCAGTTTCGAG |
| Sequence-based reagent | hCALR_Rv | Gene Design | | TGTATTCTGAGTCTCCGTGCAT |
| Cell line (*Homo sapiens*) | HEK293 cells | ATCC | CRL-1573 RRID:CVCL_0045 | DMEM+10%FBS |
| Cell line (*Cricetulus griseus*) | CHO cells | ATCC | CCL-61 RRID:CVCL_0214 | Ham's F12+10%FBS |
| Cell line (*Homo sapiens*) | Plat-E cells | Cosmobio | RV-101 RRID:CVCL_B488 | Ecotropic retrovirus packaging DMEM+10%FBS |
| Cell line (*Mus musculus*) | C2C12 cells | RIKEN cell bank | RCB0987 RRID:CVCL_0188 | C2C12 sleletal myoblast DMEM+10%FBS |
| Recomninant DNA reagent | mCdh13 | This paper | | Materials and methods: plasmids |
| Recomninant DNA reagent | mCalr | This paper | | Materials and methods: plasmids |
| Recomninant DNA reagent | mAdipor1 | This paper | | Materials and methods: plasmids |
| Recomninant DNA reagent | hCDH13 | This paper | | Materials and methods: plasmids |
| Recomninant DNA reagent | hCALR | This paper | | Materials and methods: plasmids |
| Recomninant DNA reagent | hADIPOR1 | This paper | | Materials and methods: plasmids |
| Recomninant DNA reagent | mCat1 | This paper | | Materials and methods: plasmids |

## Plasmids

General PCR techniques were used for the construction of plasmids. All primers were purchased from GeneDesign, Inc. The full-length cDNAs of human and mouse T-cadherin (*mCdh13*), AdipoR1 (*mAdipor1*), and calreticulin (*mCalr*) were cloned into pcDNA mammalian expression plasmid vectors. The PA tag sequence (GVAMPGAEDDVV) was attached to the N termini of mouse T-cadherin and calreticulin and the C terminus of AdipoR1. Mouse *mCat1* cDNA was cloned into a pcDNA mammalian expression plasmid vector.

## Cell lines

Mammalian cell lines were obtained from the American Type Culture Collection or RIKEN BRC CELL BANK. All cell lines negative for mycoplasma contamination were maintained under conditions indicated in Key resources table.

## Stably expressing CHO cells

PA-tagged receptor cDNAs were subcloned into the retrovirus packaging vector pMXs-neo, and the resultant vectors were used to transfect Plat-E cells, thereby generating recombinant retroviruses. CHO cells were transfected with a mouse *mCat* plasmid, and after 48 hr, the resultant cells were infected with recombinant retroviruses. G418 at 800 µg/mL was used to select stably introduced cells.

## Antibodies

The following primary antibodies were used: goat polyclonal anti-mouse adiponectin (AF1119, R and D), goat polyclonal anti-human adiponectin (AF1065, R and D), goat polyclonal anti-T-cadherin (AF3264, R and D), rabbit monoclonal anti-α-tubulin (11H10, Cell Signaling), rat monoclonal anti-PA-tag (human podoplanin PLAG sequence) (012–25863, FUJIFILM), and rabbit monoclonal anti-GPADH (14C10, Cell Signaling Technology).

## Animal

Mouse serum was obtained from male and female C57BL6J jcl mice. Mice were maintained at 22°C under a 12:12 hr light-dark cycle (lights on from 8:00 AM to 8:00 PM). The experimental protocol was approved as No. 28-072-023 by the Ethics Review Committee for Animal Experimentation of Osaka University School of Medicine. This study also conformed to the Guide for the Care and Use of Laboratory Animals published by the US National Institutes of Health.

## Binding study

Adiponectin binding studies were performed using serum as the source of adiponectin in situ without any processing. Cells were treated with the indicated concentrations of serum in serum-free DMEM at 4°C for 1 hr and then washed with serum-free DMEM three times. NZ-1 binding was performed by incubating cells with 1.0 µg/mL NZ-1 in DMEM containing 0.2%BSA at 4°C for 1 hr and washed with serum-free DMEM three times. Cell lysates were combined with Laemmli sample buffer for SDS-PAGE and heated at 98°C for 5 min or combined with native-page buffer (*Suzuki et al., 2007*).

## Adiponectin concentration

Adiponectin concentrations in sample preparations were measured by ELISA (Otsuka Pharmaceutical Co.).

## Western blotting

Whole cell lysates were loaded onto 4–20% gradient SDS-PAGE gels (Bio-Rad) and transferred to nitrocellulose membranes. Membranes were blocked with PVDF Blocking Reagent for the Can Get signal (TOYOBO), incubated with primary antibodies using Can Get signal solution 1 (TOYOBO) at 4°C overnight, and then incubated with secondary antibodies conjugated with HRP using Can Get signal solution 2 (TOYOBO) at room temperature (RT) for 60 min. Chemiluminescence signals developed with Chemi-Lumi One Super (Nacalai Tesque) were visualized by ChemiDoc Touch and quantitated using Image Lab software (Bio-Rad). A native-PAGE analysis of the multimer composition of adiponectin was performed according to the method described (*Suzuki et al., 2007*).

## Cell surface protein biotinylation

Cell surface protein biotinylation and subsequent isolation were performed using the Cell Surface Biotinylation Kit (Pierce) according to the instructions provided by the manufacturer.

## Immunofluorescence staining

Cells on coverslips were fixed with periodate-lysine-paraformaldehyde (PLP) for 30 min and incubated with 3% w/v BSA and 0.3% w/v Triton X-100 in Dulbecco's phosphate-buffered saline without calcium or magnesium (PBS) for 60 min. Cells were then incubated with 10 µg/mL NZ-1 at 4°C overnight and then incubated with an Alexa-Fluor 488 secondary antibody at RT for 60 min. Cell nuclei were counterstained with DAPI. A microscopy analysis was performed using an Olympus FV1000D confocal laser scanning microscope system (Olympus).

## Statistical analysis

Values were expressed as the mean $\pm$ SEM. Differences between variables were compared using the Student's t-test. The probability (P) values of $<0.05$ were considered to be significant.

## Data and software availability

All data were deposited in Dryad at https://doi.org/10.5061/dryad.82557c0.

## Acknowledgements

The authors thank the staff of the Center of Medical Research and Education, Graduate School of Medicine Osaka University, for their excellent technical support. This work was supported in part by a Grant-in-Aid for Scientific Research (C) #16K09802 (to SK), a Grant-in-Aid for Scientific Research (C) #16K09801 (to NM), and a Grant-in-Aid for Scientific Research (B) #15H04853 (to IS), the Uehara Memorial Foundation, as well as CREST and JST (to IS). The funders had no role in study design, data collection, and analysis, the decision to publish, or preparation of the manuscript.

## Additional information

### Funding

| Funder | Grant reference number | Author |
|---|---|---|
| Japan Science and Technology Agency | #16K09802 | Shunbun Kita |
| Japan Science and Technology Agency | #16K09801 | Norikazu Maeda |
| Japan Science and Technology Agency | #15H04853 | Iichiro Shimomura |
| CREST | | Iichiro Shimomura |
| The Uehara Memorial Foundation | | Iichiro Shimomura |

The funders had no role in study design, data collection and interpretation, or the decision to submit the work for publication.

### Author contributions

Shunbun Kita, Conceptualization, Resources, Software, Formal analysis, Funding acquisition, Investigation, Visualization, Methodology, Writing—original draft, Writing—review and editing, Performed biochemical and cellular experiments; Shiro Fukuda, Resources, Data curation, Software, Validation, Visualization, Methodology, Writing—review and editing, Cloned the cDNAs of adiponectin receptors; Norikazu Maeda, Data curation, Supervision, Funding acquisition, Writing—review and editing; Iichiro Shimomura, Data curation, Supervision, Funding acquisition, Project administration, Writing—review and editing

### Author ORCIDs

Shunbun Kita  https://orcid.org/0000-0002-8937-0053

## Ethics

Animal experimentation: The experimental protocol was approved as No. 28-072-023 by the Ethics Review Committee for Animal Experimentation of Osaka University School of Medicine. This study also conformed to the Guide for the Care and Use of Laboratory Animals published by the US National Institutes of Health.

## Decision letter and Author response

Decision letter https://doi.org/10.7554/eLife.48675.011
Author response https://doi.org/10.7554/eLife.48675.012

# Additional files

### Supplementary files

• Transparent reporting form  DOI: https://doi.org/10.7554/eLife.48675.007

### Data availability

All data were deposited in Dryad at https://doi.org/10.5061/dryad.82557c0.

The following dataset was generated:

| Author(s) | Year | Dataset title | Dataset URL | Database and Identifier |
|---|---|---|---|---|
| Kita S, Fukuda S, Maeda N, Shimomura I | 2019 | Data from: Native adiponectin in serum binds to cells expressing T-cadherin, but not AdipoRs or calreticulin | https://dx.doi.org/10.5061/dryad.82557c0 | Dryad Digital Repository, 10.5061/dryad.82557c0 |

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
