## [Decision Letter]

[Editors’ note: this article was originally rejected after discussions between the reviewers, but the authors were invited to resubmit after an appeal against the decision.]

Thank you for submitting your work entitled "Native adiponectin existing in serum binds with T-cadherin, but not with AdipoRs nor Calreticulin" for consideration by *eLife*. Your article has been reviewed by three peer reviewers, and the evaluation has been overseen by a Reviewing Editor and a Senior Editor. The following individual involved in review of your submission has agreed to reveal their identity: Morris Birnbaum (Reviewer #3).

Our decision has been reached after consultation between the reviewers. Based on these discussions and the individual reviews below, we regret to inform you that your work will not be considered further for publication in *eLife*.

This study focused on elucidating the physiological binding partner(s) for native adiponectin. With cell culture based studies, they have revealed that serum adiponectin exclusively bound to T-cadherin, but not to the "classical" AdipoR receptors nor Calreticulin. The strength of the study is that they have shown binding of "native" adiponectin to T-cadherin as this has not been shown before. However, the study has several weaknesses which preclude publication in *eLife*. First, the binding system used is somewhat "dirty" limiting careful biochemical analyses. Second, the overall advance of the study is unclear as it has already been shown that adiponectin binds to T-cadherin. Third, all reviewers highlighted weaknesses with the binding assay pinpointing the inability of the authors to reach saturation binding, an essential criterion for a receptor-ligand interaction. Finally, as highlighted by reviewer 1, much more work is required to prove a lack of interaction between adiponectin and the other receptor systems.

Reviewer #1:

This manuscript presents data showing serum adiponectin binds to T-cadherin overexpressed in HEK293 cells as well as endogenous T-cadherin in C2C12 cells. These findings are solid and convincing since in such experiments adiponectin is in native forms and being adsorbed directly in such forms. The claim that T-cadherin is a "receptor" for adiponectin is justified by these data and conforms to other solid papers in the field. These data do not add much to the literature as it is already established that adiponectin binds to T-cadherin, although this is a new way of demonstrating the phenomenon. But the claim is convincing.

On the other hand, the failure to show binding of serum adiponectin to other purported receptor molecules (R1 and R2 as well as Calreticulin) is more difficult to interpret. The claim that these are not true receptors could be true, but it is impossible to prove a negative. What if there are "accessory" proteins required to confer binding activity to these other putative receptors? Expression of the receptors alone would not be sufficient to display binding in that case. Normally, ligand-receptor interactions are quantified by binding of pure ligand with pure receptor, which allows binding constants to be calculated. This is not the case here, since the ligand is in a complex mixture (serum) and the "receptor" is also within a complex structure (cell membrane). These complications raise concerns about the conclusion, and have caused other groups to use gene KO mice to interrogate putative "receptor" function.

On balance, this paper should be published since it raises an important question for the field. Publication would also highlight the need to determine binding affinities for the putative receptors. However, the study doesn't add mechanistic information to the field, and doesn't define why dysfunction of adiponectin occurs when the R1/R2 receptors are deleted in mice. For those reasons, it could be argued that this paper belongs in a specialty journal.

One other point is that careful editing of this manuscript for proper English usage would be mandatory prior to publication.

Reviewer #2:

In this study, Kita et al. have focused on the elucidation of physiological binding partner(s) for native adiponectin. With cell culture based studies, they have revealed that serum adiponectin exclusively bound to T-cadherin, but not to AdipoR nor Calreticulin. While this study contains potential interests, several issues need to be properly addressed to enhance the significance of this study.

1) Due to molecular structure of T-cadherin protein, it remains elusive to guess the mode(s) of action for adiponectin signaling. More discussion and/or speculation would be more helpful for general readers to understand adiponectin action and signaling.

2) Along the same line, it is unclear whether T-cadherin might behave as a typical receptor protein for adiponectin. For example, typical receptors are saturated by ligands. However, provided data (Figures 1, 2 and 3) did not show any saturation pattern nor dose response. This issue needs to be properly addressed by experimental data.

3) The authors demonstrated that native adiponectin bound to T-cadherin. Have you compared the binding affinity with native and recombinant adiponectin proteins to three potential receptors? If there is any difference, it needs to be described and discussed.

Reviewer #3:

In spite of the identification of adiponectin quite a number of years ago, the nature of its receptors remains controversial. The rationale for the current study is that the major criterion for evaluating a potential receptor should be that it binds native adiponectin and not just recombinant product. This seems quite reasonable and therefore the strength of the paper is that it seeks to accomplish that. The weakness is that the nature of the protocol involves performing the binding study on a very complex, on might say "dirty" system, in which it is challenging to infer the actual biochemical even being measured. At a minimum, the authors' need to be much more precise in their language and judicious in what they claim the data show. For example, the authors state "These results clearly indicate that T-cadherin can bind native adiponectin." This is a gross overinterpretation of this experiment. What the authors show is that overexpression of T-cadherin in cells allows them to bind increased amounts of adiponectin. There is no information to indicated that the adiponectin is binding directly to T-cadherin or, for that matter, that T-cadherin is involved in the binding of hormone. This interpretation might be plausible, but it is far from proven.

The authors do demonstrate that the binding is concentration dependent, which is important for true receptor binding, but they do not establish saturability, which is essential to arguing the physiological relevance of the event being measured. Thus, it would add to the weight of the argument of they could show a full binding curve up to the point of saturability. It is relatively straightforward to vary expression to make sure the receptor level is low enough that it can be saturated with the adiponectin levels present in serum.

[Editors’ note: what now follows is the decision letter after the authors submitted for further consideration.]

Congratulations, we are pleased to inform you that your article, "Native adiponectin in serum binds to cells expressing T-cadherin, but not AdipoRs or calreticulin", has been accepted for publication in *eLife* as a Short Report article.

While the reviewers still identified some shortcomings with your study, it was generally considered that your paper makes an important contribution to the field. In particular, there is an abundance of evidence in the literature to show that adiponectin signals via the ADIPOR1 and 2 receptors. Your work questions the breadth of this conclusion and shows that as a minimum the field needs to carefully consider the role of this alternate adiponectin receptor, T cadherin. This paper clearly shows that adiponectin binds to T-cadherin with considerable affinity under somewhat physiological conditions. As to whether the origin or nature of the ligand influences the binding specificity is unclear but this is certainly something important to consider that this work highlights. While, as pointed out by two of the reviewers, there is a long way to go to ascertain how the adiponectin-T-cadherin interaction signals in cells this is perhaps something for a future study.

Please address as best you can each of the points raised by the reviewers as noted below. Also I noted that some restructuring of your figures is necessary to consolidate space and to improve the presentation of the data. Furthermore, I still believe you could shorten the article without loss of clarity.

Reviewer #1:

This paper provides data supporting the view that binding of adiponectin to cells is through T-cadherin rather than other purported receptor types. There are three issues that diminish the impact of the paper:

1) The first paragraph is an overly long "stream of consciousness" type of Introduction that should be heavily edited and organized into multiple paragraphs around the several discrete ideas.

2) The major impact of this study would be that bio-effects of adiponectin are mediated through T-cadherin, not the other putative receptors, but unfortunately no bio-effects are studied in these experiments. While the results are very interesting and certainly provocative, a definitive experiment on this key point showing that T-cadherin but not the other receptors mediates important bio-effects is missing.

3) The cell biology/microscopy data are too limited. Multiple fields should be shown and appropriate controls with null signal. The data should be quantified from many fields as well.

Reviewer #2:

In this work, Kita et al. have demonstrated that native adiponectin would exclusively bind to T-cadherin. With cell culture model, they showed that serum or purified adiponectin preferentially bound to T-cadherin but not to AdipoR nor Calreticulin. Although they provided more data and tried to strengthen Discussion part in the resubmitted manuscript, it still has technical limitations to reach the firm conclusion.

1) Although it has been reported that adiponectin binds to AdipoRs, calreticulin, and T-cadherin, they argued that the major binding protein for native adiponectin is T-cadherin. If the binding affinity of native adiponectin to adipoRs or calreticulin would be much weaker than T-cadherin, it seems that the experimental condition in this study might be too harsh to detect weak interaction with others. They have to adopt alternative methods to affirm the extent of protein interactions. Along the same line, they need to measure/compare binding affinity of recombinant adiponectin with other proteins including AdipoR and Calreticulin.

2) To convince that native adiponectin exclusively binds to T-cadherin, it is crucial to provide positive control data.

3) It has been suggested to examine intracellular signaling cascades such as phospho-AMPK and phospho-p38 upon the interaction between native adiponectin and potential receptors.

Reviewer #3:

I was ambivalent about this manuscript the first time around and the authors have only improved it with the additional experiments. In all honesty, this will never be the cleanest study and will not have the highest level of impact due to lack of mechanism, whatever the authors do to address the issues. However, I continue to believe that it brings up a valid point about the ambiguity of the true adiponectin receptor and the data are strong enough to provide a valid challenge to the dogma. On balance, I recommend acceptance.

---

## [Author Response]

[Editors’ note: the author responses to the first round of peer review follow.]

This study focused on elucidating the physiological binding partner(s) for native adiponectin. With cell culture based studies, they have revealed that serum adiponectin exclusively bound to T-cadherin, but not to the "classical" AdipoR receptors nor Calreticulin. The strength of the study is that they have shown binding of "native" adiponectin to T-cadherin as this has not been shown before.However, the study has several weaknesses which preclude publication in eLife. First, the binding system used is somewhat "dirty" limiting careful biochemical analyses.

We used HMW adiponectin purified from serum (Fukuda et al., 2017, PMID: 28325833) and full-length recombinant adiponectin produced by HEK293 cells in our revised manuscript, and they gave essentially the same results. Furthermore, we would like to emphasize that the ligand in serum is the most natural ligand that maintains its ability to interact with the physiological binding partner. Numerous constituents, including a large number of proteins, exert adequate blocking effects to investigate whether the interaction between the physiological ligand and binding protein is specific. Moreover, the use of a recombinant ligand may make the experiment unreproducible because the same recombinant ligand cannot be made at another site. The ligand-receptor interaction in a more purified system can have mean if the ligand can bind with the receptor in a more physiological situation like in serum as our experiment. The results of our new experiment indicate that recombinant adiponectin has a limited affinity to T-cadherin due to the presence of less of the HMW form.

The results obtained were shown in Figure 1B and 1D. We added the following sentences.

Results and Discussion

“We directly examined mouse serum as the ligand solution, including the most native adiponectin, purified adiponectin from mouse serum (Fukuda et al., 2017), and full-length recombinant adiponectin produced in HEK293 cells. […] Recombinant adiponectin contained a lower amount of HMW multimer adiponectin than mouse serum and purified adiponectin from serum (Figure 1B).

Results and Discussion

“The treatment of cells with different preparations of adiponectin at 4°C for 1 hr resulted in the binding of prepared adiponectin only to cells expressing mouse T-cadherin (Figure 1D). Mouse serum and purified adiponectin showed similar binding, whereas recombinant adiponectin containing a lower amount of 6-mer and the HMW multimer exhibited markedly weaker binding (Figure 1D).”

Second, the overall advance of the study is unclear as it has already been shown that adiponectin binds to T-cadherin.

The binding of HMW recombinant adiponectin to T-cadherin was initially discovered in 2004 (Hug C et al., 2004); however, it has not been investigated in detail because it lacks an intracellular domain. We previously reported a molecular interaction between serum derived purified HMW adiponectin and Fc-fusion purified T-cadherin with high affinity (KD=1.0 nM) (Fukuda et al., 2017). Moreover, we demonstrated that T-cadherin mediated the exosome-stimulating function of adiponectin (Obata et al., 2018). However, as discussed in the Abstract, reported receptors for adiponectin have not yet been simultaneously examined on their adiponectin binding under equal conditions. The present study established by WB that native HMW (> 6mer) adiponectin in serum and highly purified serum-derived multimer adiponectin bound to cells expressing T-cadherin, but not to those expressing AdipoRs or calreticulin. An important issue in the present study is that native HMW adiponectin, either in serum or highly purified, did not appear to bind to cells expressing AdipoRs or calreticulin, but did bind to those expressing T-cadherin in the same set of experiments. Based on these results, we speculate that AdipoRs and/or calreticulin may function as adiponectin receptors for LMW forms such as globular adiponectin, possibly on the site of cells after HMW adiponectin binding to cell-surface T-cadherin is cleaved. We discussed this possibility in our response to comment 4 by the Editor.

Third, all reviewers highlighted weaknesses with the binding assay pinpointing the inability of the authors to reach saturation binding, an essential criterion for a receptor-ligand interaction.

We showed the saturable binding of native adiponectin in serum in Figure 1E. The binding isotherm obtained is consistent with the high-affinity interaction we reported previously using purified HMW adiponectin with purified T-cadherin (Fukuda et al., 2017).

The results obtained were shown in Figure 1E. We added the following sentence.

Results and Discussion

“The dose-response study revealed the specific and saturable binding of native adiponectin in serum to cells expressing T-cadherin (Figure 1E).”

Finally, as highlighted by reviewer 1, much more work is required to prove a lack of interaction between adiponectin and the other receptor systems.

We simultaneously compared the candidate receptors. The expression of T-cadherin gave a strong band of bound adiponectin, which is consistent with our previous finding showing that recombinant T-cadherin binds HMW adiponectin in a 1: 1 ratio with high affinity. If the receptor has meaningful affinity, it will bind and give a visible band on WB, similar to that of T-cadherin. However, there was no detectable band on the AdipoR or calreticulin lane. Furthermore, AdipoR was discovered by expression cloning. Therefore, the ectopic introduction of AdipoRs promotes adiponectin binding. The Nature study also showed that HEK293 cells overexpressing AdipoRs bound *E. coli* recombinant globular adiponectin (Supplementary Figure 2A and 2B, Yamauchi et al., 2003). Therefore, difficulties are associated with speculating about the requirement for “accessory” proteins to confer adiponectin binding activity to AdipoRs.

Numerous studies have indicated that AdipoRs mediate adiponectin signaling. Therefore, there may be additional activating mechanisms of AdipoRs by adiponectin, i.e., reductive or proteolytic generation of the trimer, monomer, and/or globular adiponectin. The present study focused on the direct binding of native HMW adiponectin (more than a hexamer) and the results obtained suggest the activation of AdipoRs by these LMW forms, possibly on the site of cells after HMW adiponectin binding to cell surface T-cadherin is cleaved.

On the other hand, in the case of calreticulin, this was evidenced by a neutralizing antibody treatment inhibiting the adiponectin interaction with cells (Takemura et al., 2007). Therefore, “accessory” proteins may be required to confer adiponectin binding activity to calreticulin. We also revised our discussion on this point to keep the possibility of calreticulin mediating the adiponectin interaction with cells.

We added a detailed discussion on this important assumption as follows.

Results and Discussion

“Adiponectin is an atypically abundant circulating factor that is exclusively secreted from adipocytes as a trimer, hexamer, and HMW multimer. […] Therefore, some “accessory” proteins may be required to confer adiponectin binding activity to calreticulin.”

We also revised the Abstract as follows.

“Adiponectin is an adipocyte-derived atypically abundant circulating factor that protects various organs and tissues through its receptors, AdipoRs, calreticulin, and T-cadherin. […] Collectively, these results suggest that T-cadherin is the major binding partner of native adiponectin in serum.”

We also revised conclusive sentences at the bottom of the Results and Discussion sections as follows.

“The present results may further contribute to clarifying the activating mechanism of AdipoRs by LMW adiponectin, generated by reduction or proteolytic cleavage, followed by HMW adiponectin binding to cell surface T-cadherin.”

Reviewer #1:This manuscript presents data showing serum adiponectin binds to T-cadherin overexpressed in HEK293 cells as well as endogenous T-cadherin in C2C12 cells. These findings are solid and convincing since in such experiments adiponectin is in native forms and being adsorbed directly in such forms. The claim that T-cadherin is a "receptor" for adiponectin is justified by these data and conforms to other solid papers in the field. These data do not add much to the literature as it is already established that adiponectin binds to T-cadherin, although this is a new way of demonstrating the phenomenon. But the claim is convincing.

The binding of HMW recombinant adiponectin to T-cadherin was initially discovered in 2004 (Hug C et al., 2004); however, it has not been investigated in detail because it lacks an intracellular domain. We previously reported a molecular interaction between serum-derived purified HMW adiponectin and Fc-fusion purified T-cadherin with high affinity (KD=1.0 nM) (Fukuda et al., 2017). Moreover, we demonstrated that T-cadherin mediated the exosome-stimulating function of adiponectin (Obata et al., 2018). However, as discussed in the Abstract, reported receptors for adiponectin have not yet been simultaneously examined under equal conditions on their adiponectin binding. The present study established by WB that native HMW (> 6mer) adiponectin in serum and highly purified serum derived multimer adiponectin bound to cells expressing T-cadherin, but not to those expressing AdipoRs or calreticulin. An important issue in the present study is that native HMW adiponectin, either in serum or highly purified, did not appear to bind to cells expressing AdipoRs or calreticulin, but did bind to those expressing T-cadherin in the same set of experiments. Based on these results, we speculate that AdipoRs and/or calreticulin may function as adiponectin receptors for LMW forms such as globular adiponectin, possibly on the site of cells after HMW adiponectin binding to cell surface T-cadherin is cleaved. We discussed this possibility in our response to reviewer #1’s comment 3.

On the other hand, the failure to show binding of serum adiponectin to other purported receptor molecules (R1 and R2 as well as Calreticulin) is more difficult to interpret. The claim that these are not true receptors could be true, but it is impossible to prove a negative. What if there are "accessory" proteins required to confer binding activity to these other putative receptors? Expression of the receptors alone would not be sufficient to display binding in that case. Normally, ligand-receptor interactions are quantified by binding of pure ligand with pure receptor, which allows binding constants to be calculated. This is not the case here, since the ligand is in a complex mixture (serum) and the "receptor" is also within a complex structure (cell membrane). These complications raise concerns about the conclusion, and have caused other groups to use gene KO mice to interrogate putative "receptor" function.

We appreciate the valuable comment. In the revised manuscript, we compared the binding of native adiponectin in serum, purified native adiponectin, and recombinant full-length adiponectin produced in HEK293 cells. The results obtained indicated that only cells expressing T-cadherin and not the other receptors bound these preparations of adiponectin. However, recombinant adiponectin with a lower amount of HMW multimer adiponectin showed weak binding at a physiological concentration (20 µg/mL) of adiponectin. The results obtained were shown in Figure 1B and 1D. We added the following sentences.

Results and Discussion

“We directly examined mouse serum as the ligand solution, including the most native adiponectin, purified adiponectin from mouse serum (Fukuda et al., 2017), and full-length recombinant adiponectin produced in HEK293 cells. […] Recombinant adiponectin contained a lower amount of HMW multimer adiponectin than mouse serum and purified adiponectin from serum (Figure 1B).”

Results and Discussion

“The treatment of cells with different preparations of adiponectin at 4°C for 1 hr resulted in the binding of prepared adiponectin only to cells expressing mouse T-cadherin (Figure 1D). Mouse serum and purified adiponectin showed similar binding, whereas recombinant adiponectin containing a lower amount of 6-mer and the HMW multimer exhibited markedly weaker binding (Figure 1D).”

Furthermore, we would like to emphasize that the ligand in serum is the most natural ligand that maintains its ability to interact with the physiological binding partner. Numerous constituents, including a large number of proteins, exert adequate blocking effects to investigate whether the interaction between the physiological ligand and binding protein is specific. Moreover, the use of a recombinant ligand may make the experiment unreproducible because the same recombinant ligand cannot be made at another site. The ligand-receptor interaction in a more purified system can have mean if the ligand can bind with the receptor in a more physiological situation like in serum as our experiment The results of our new experiment indicate that recombinant adiponectin has a limited affinity to T-cadherin due to the presence of less of the HMW form.

The expression of T-cadherin gave a strong band of bound adiponectin, which is consistent with our previous finding showing that recombinant T-cadherin binds HMW adiponectin in a 1: 1 ratio with high affinity. If the receptor has meaningful affinity, it will bind and give a visible band on WB, similar to that of T-cadherin. However, there was no detectable band on the AdipoR or calreticulin lane. Furthermore, AdipoR was discovered by expression cloning. Therefore, the ectopic introduction of AdipoRs promotes adiponectin binding. The Nature study also showed that HEK293 cells overexpressing AdipoRs bound *E. coli* recombinant globular adiponectin (Supplementary Figure 2A and 2B, Yamauchi et al., 2003). Therefore, difficulties are associated with speculating about the requirement for “accessory” proteins to confer adiponectin binding activity to AdipoRs.

On the other hand, in the case of calreticulin, this was evidenced by a neutralizing antibody treatment inhibiting the adiponectin interaction with cells (Takemura Y et al., 2007). Therefore, “accessory” proteins may be required to confer adiponectin binding activity to calreticulin. We also revised our discussion on this point to keep the possibility of calreticulin mediating the adiponectin interaction with cells.

We added a detailed discussion on this important assumption as follows.

Results and Discussion

“We simultaneously compared three adiponectin receptors. The expression of T-cadherin gave a strong band of bound adiponectin, which is consistent with our previous finding showing that recombinant Tcadherin binds HMW adiponectin in a 1: 1 ratio with high affinity (Fukuda et al., 2017). […] Therefore, difficulties are associated with speculating about the requirement for some “accessory” proteins to confer adiponectin binding activity to AdipoRs.”

Results and Discussion

“On the other hand, in the case of calreticulin, this was evidenced by a neutralizing antibody treatment inhibiting the adiponectin interaction with cells (Takemura et al., 2007). Therefore, some “accessory” proteins may be required to confer adiponectin binding activity to calreticulin.”

On balance, this paper should be published since it raises an important question for the field. Publication would also highlight the need to determine binding affinities for the putative receptors. However, the study doesn't add mechanistic information to the field, and doesn't define why dysfunction of adiponectin occurs when the R1/R2 receptors are deleted in mice. For those reasons, it could be argued that this paper belongs in a specialty journal.

We appreciate the valuable comment. Numerous studies indicated that AdipoRs mediate adiponectin signaling. Therefore, there may be additional activating mechanisms of AdipoRs by adiponectin, i.e., reductive or proteolytic generation of the trimer, monomer, and/or globular adiponectin. The present study focused on the direct binding of native HMW adiponectin (more than hexamer) and the results obtained suggest the activation of AdipoRs by these LMW forms, possibly on the site of cells after HMW adiponectin binding to cell-surface T-cadherin is cleaved. We added a detailed discussion on this important assumption as follows.

Results and Discussion

“Since numerous studies indicated that AdipoRs mediate adiponectin signaling in a number of cell types, an additional activating mechanism for AdipoRs by adiponectin, i.e., the reductive or proteolytic generation of the trimer, monomer, and/or globular adiponectin, may exist. The results of the present study, which focused on the direct binding of native HMW adiponectin, may indicate the activation of AdipoRs by low-molecular-weight (LMW) forms.”

We also revised the Abstract as follows.

“Adiponectin is an adipocyte-derived atypically abundant circulating factor that protects various organs and tissues through its receptors, AdipoRs, calreticulin, and T-cadherin. […] Collectively, these results suggest that T-cadherin is the major binding partner of native adiponectin in serum.”

We also revised conclusive sentences at the bottom of the Results and Discussion sections as follows.

“The present results may further contribute to clarifying the activating mechanism of AdipoRs by LMW adiponectin, generated by reduction or proteolytic cleavage, followed by HMW adiponectin binding to cell surface T-cadherin.”

One other point is that careful editing of this manuscript for proper English usage would be mandatory prior to publication.

The revised manuscript was proofread by a native speaker of English.

Reviewer #2:In this study, Kita et al. have focused on the elucidation of physiological binding partner(s) for native adiponectin. With cell culture based studies, they have revealed that serum adiponectin exclusively bound to T-cadherin, but not to AdipoR nor Calreticulin. While this study contains potential interests, several issues need to be properly addressed to enhance the significance of this study.1) Due to molecular structure of T-cadherin protein, it remains elusive to guess the mode(s) of action for adiponectin signaling. More discussion and/or speculation would be more helpful for general readers to understand adiponectin action and signaling.

We appreciate the valuable comment. We added a detailed discussion on the mode of action of adiponectin through T-cadherin as below.

Results and Discussion

“T-cadherin binds clinically important HMW multimer adiponectin with high affinity (Fukuda et al., 2017) and mediates adiponectin-induced exosome biogenesis and ceramide efflux to exosomes (Obata et al., 2018). […] These findings support Tcadherin mediating adiponectin functions as a receptor for native HMW multimer adiponectin (Kita et al., 2019).”

2) Along the same line, it is unclear whether T-cadherin might behave as a typical receptor protein for adiponectin. For example, typical receptors are saturated by ligands. However, provided data (Figures 1, 2 and 3) did not show any saturation pattern nor dose response. This issue needs to be properly addressed by experimental data.

We appreciate the valuable comment. We demonstrated the saturable binding of native adiponectin in serum. The binding isotherm obtained is consistent with the high-affinity interaction we reported previously using purified HMW adiponectin with purified T-cadherin (Fukuda et al., 2017). The results obtained were shown in Figure 1D. The following sentences were added.

Results and Discussion

“The dose-response study revealed the specific and saturable binding of native adiponectin in serum to cells expressing T-cadherin (Figure 1E).”

3) The authors demonstrated that native adiponectin bound to T-cadherin. Have you compared the binding affinity with native and recombinant adiponectin proteins to three potential receptors? If there is any difference, it needs to be described and discussed.

We appreciate the valuable comment. In the revised manuscript, we compared the binding of native adiponectin in serum, purified native adiponectin, and recombinant full-length adiponectin produced in HEK293 cells. The results obtained indicated that only cells expressing T-cadherin and not the other receptors bound these preparations of adiponectin. However, recombinant adiponectin with a lower amount of HMW multimer adiponectin showed weak binding at a physiological concentration (20 μg/mL) of adiponectin.

The results obtained were shown in Figure 1B and 1D. The following sentences were added.

Results and Discussion

“We directly examined mouse serum as the ligand solution, including the most native adiponectin, purified adiponectin from mouse serum (Fukuda et al., 2017), and full-length recombinant adiponectin produced in HEK293 cells. […] Recombinant adiponectin contained a lower amount of HMW multimer adiponectin than mouse serum and purified adiponectin from serum (Figure 1B).”

Results and Discussion

“The treatment of cells with different preparations of adiponectin at 4°C for 1 hr resulted in the binding of prepared adiponectin only to cells expressing mouse T-cadherin (Figure 1D). Mouse serum and purified adiponectin showed similar binding, whereas recombinant adiponectin containing a lower amount of 6-mer and the HMW multimer exhibited markedly weaker binding (Figure 1D).”

Once again, we are sincerely grateful to reviewer #2 for the constructive suggestions, which have improved the quality of the revised manuscript.

Reviewer #3:In spite of the identification of adiponectin quite a number of years ago, the nature of its receptors remains controversial. The rationale for the current study is that the major criterion for evaluating a potential receptor should be that it binds native adiponectin and not just recombinant product. This seems quite reasonable and therefore the strength of the paper is that it seeks to accomplish that. The weakness is that the nature of the protocol involves performing the binding study on a very complex, on might say "dirty" system, in which it is challenging to infer the actual biochemical even being measured.

We appreciate the valuable comment. We compared the binding of native adiponectin in serum, purified native adiponectin, and recombinant full-length adiponectin produced in HEK293 cells. The results obtained indicated that only cells expressing T-cadherin and not the other receptors bound these preparations of adiponectin. However, recombinant adiponectin with lower amounts of HMW multimer adiponectin showed weak binding at a physiological concentration (20 μg/mL) of adiponectin. These results were shown in Figure 1B and 1D. We added the following sentences.

Results and Discussion

“We directly examined mouse serum as the ligand solution, including the most native adiponectin, purified adiponectin from mouse serum (Fukuda et al., 2017), and full-length recombinant adiponectin produced in HEK293 cells. […] Recombinant adiponectin contained a lower amount of HMW multimer adiponectin than mouse serum and purified adiponectin from serum (Figure 1B).”

Results and Discussion

“The treatment of cells with different preparations of adiponectin at 4°C for 1 hr resulted in the binding of prepared adiponectin only to cells expressing mouse T-cadherin (Figure 1D). Mouse serum and purified adiponectin showed similar binding, whereas recombinant adiponectin containing a lower amount of 6-mer and the HMW multimer exhibited markedly weaker binding (Figure 1D).”

Furthermore, we would like to emphasize that the ligand in serum is the most natural ligand that maintains its ability to interact with the physiological binding partner. Numerous constituents, including a large number of proteins, exert adequate blocking effects to investigate whether the interaction between the physiological ligand and binding protein is specific. Moreover, the use of a recombinant ligand may make the experiment unreproducible because the same recombinant ligand cannot be made at another site. The ligand-receptor interaction in a more purified system can have mean if the ligand can bind with the receptor in a more physiological situation like in serum as our experiment The results of our new experiment indicate that recombinant adiponectin has limited affinity to T-cadherin due to the presence of less of the HMW form.

At a minimum, the authors' need to be much more precise in their language and judicious in what they claim the data show. For example, the authors state "These results clearly indicate that T-cadherin can bind native adiponectin." This is a gross overinterpretation of this experiment. What the authors show is that overexpression of T-cadherin in cells allows them to bind increased amounts of adiponectin. There is no information to indicated that the adiponectin is binding directly to T-cadherin or, for that matter, that T-cadherin is involved in the binding of hormone. This interpretation might be plausible, but it is far from proven.

We appreciate the valuable comment. The revised manuscript was proofread by a native English speaker.

Since we previously reported a serum-derived purified APN interaction with purified Fc-fusion Tcadherin (JBC, 2017), the statement “These results clearly indicate that T-cadherin can bind native adiponectin”was replaced as follows.

Results and Discussion

“We previously reported that purified recombinant T-cadherin bound purified adiponectin with an affinity of KD=1.0 nM (Fukuda et al., 2017). The present results showed the saturable binding of native adiponectin in serum to cells expressing T-cadherin, which is consistent with previous findings (Fukuda et al., 2017).”

The authors do demonstrate that the binding is concentration dependent, which is important for true receptor binding, but they do not establish saturability, which is essential to arguing the physiological relevance of the event being measured. Thus, it would add to the weight of the argument of they could show a full binding curve up to the point of saturability. It is relatively straightforward to vary expression to make sure the receptor level is low enough that it can be saturated with the adiponectin levels present in serum.

We appreciate the valuable comment. We demonstrated the saturable binding of native adiponectin in serum. The binding isotherm obtained is consistent with the high-affinity interaction we reported previously using purified HMW adiponectin with purified T-cadherin (Fukuda et al., 2017). The results obtained were shown in Figure 1E. The following sentences were added.

Results and Discussion

“The dose-response study revealed the specific and saturable binding of native adiponectin in serum to cells expressing T-cadherin (Figure 1E).”

[Editors’ note: the author responses to the re-review follow.]

Reviewer #1:This paper provides data supporting the view that binding of adiponectin to cells is through T-cadherin rather than other purported receptor types. There are three issues that diminish the impact of the paper:1) The first paragraph is an overly long "stream of consciousness" type of Introduction that should be heavily edited and organized into multiple paragraphs around the several discrete ideas.

We reorganized Introduction section according to your kind advice as follows.

“Adiponectin is a circulating factor that is secreted from adipocytes (Hu et al., 1996; Maeda et al., 1996; Nakano et al., 1996; Scherer et al., 1995). […] We herein demonstrated that native adiponectin in serum bound to cells expressing T-cadherin, but not to those expressing AdipoRs or calreticulin.”

2) The major impact of this study would be that bio-effects of adiponectin are mediated through T-cadherin, not the other putative receptors, but unfortunately no bio-effects are studied in these experiments. While the results are very interesting and certainly provocative, a definitive experiment on this key point showing that T-cadherin but not the other receptors mediates important bio-effects is missing.

Thank you for pointing the most important point to be addressed. It is well known that adiponectin stimulates intracellular signaling cascades such as phospho-AMPK and phospho-p38. And the loss of AdipoRs was shown to decrease these signaling. One explanation will be the generation of low molecular weight form of adiponectin such as globular form after binding to T-cadherin. However, it was recently reported that the primary cellular function of AdipoR1 and AdipoR2 is to maintain membrane fluidity (Ruiz M et al., J Lipid Res. 2019 May;60(5):995-1004). Loss of them might disturb membrane fluidity and thereby decrease signaling. Another explanation may be intracellular signaling by HMW-adiponectin internalization through binding to T-cadherin without interaction with AdipoRs. In future studies, we will carefully examine whether HMW-adiponectin stimulates such signaling or not and whether endocytosis of adiponectin through T-cadherin is required or not.

3) The cell biology/microscopy data are too limited. Multiple fields should be shown and appropriate controls with null signal. The data should be quantified from many fields as well.

We appreciate the valuable comment. The fluorescent microscopy data (Figure 2H) only indicate that the tagged receptors are correctly expressed, together with Western blotting data (Figure 2E). At this moment, it seems kind of reasonable to us that quantification by microscopy data may not be required because we have more accurate quantified data by Western blots.

Reviewer #2:In this work, Kita et al. have demonstrated that native adiponectin would exclusively bind to T-cadherin. With cell culture model, they showed that serum or purified adiponectin preferentially bound to T-cadherin but not to AdipoR nor Calreticulin. Although they provided more data and tried to strengthen Discussion part in the resubmitted manuscript, it still has technical limitations to reach the firm conclusion.1) Although it has been reported that adiponectin binds to AdipoRs, calreticulin, and T-cadherin, they argued that the major binding protein for native adiponectin is T-cadherin. If the binding affinity of native adiponectin to adipoRs or calreticulin would be much weaker than T-cadherin, it seems that the experimental condition in this study might be too harsh to detect weak interaction with others. They have to adopt alternative methods to affirm the extent of protein interactions. Along the same line, they need to measure/compare binding affinity of recombinant adiponectin with other proteins including AdipoR and Calreticulin.

Thank you for pointing an important discussion point in this study. As discussed in the Results and Discussion section (ninth paragraph), in the case of calreticulin, the interaction with adiponectin was evidenced by a neutralizing antibody treatment inhibiting the adiponectin interaction with cells expressing calreticulin (Takemura et al., 2007). So, possibilities of the requirement of calreticulin for adiponectin binding such as the requirement of some accessory protein may exist. In the case of AdipoRs, AdipoR was discovered by expression cloning, as discussed in the Results and Discussion section (eighth paragraph). Expression cloning requires repeated washing steps before separation by FACS. Therefore, the overexpression of AdipoR is expected to promote firm ligand binding. The initial discovery of AdipoRs also indicated that HEK293 cells overexpressing AdipoRs bound *E. coli* recombinant globular adiponectin (Yamauchi et al., 2003). Therefore, difficulties are associated with speculating about the much weaker binding affinity or requirement for some “accessory” proteins to confer adiponectin binding activity to AdipoRs. Rather, HMW-adiponectin may not bind to AdipoRs, but the globular form may do.

2) To convince that native adiponectin exclusively binds to T-cadherin, it is crucial to provide positive control data.

We appreciate the valuable comment. We previously reported and established HMW-adiponectin binding to T-cadherin in several ways including Biacore analysis and loss of tissue accumulation of adiponectin in T-cadherin KO mice (Fukuda et al., 2017, Matsuda et al., 2015). Therefore in this sense, T-cadherin can be thought of as positive control. Based on nearly the same amounts of receptors expressed on the cell surface as judged by PA-tag expression outside of the cells, native adiponectin bound to the cells expressing T-cadherin but not to the cells expressing AdipoR1.

3) It has been suggested to examine intracellular signaling cascades such as phospho-AMPK and phospho-p38 upon the interaction between native adiponectin and potential receptors.

Thank you for pointing the most important point to be addressed. It has been shown that adiponectin stimulates intracellular signaling cascades such as phospho-AMPK and phospho-p38. And the loss of AdipoRs decreases these signaling. One explanation will be the generation of low molecular weight form of adiponectin such as globular form after binding to T-cadherin. However, it was recently reported that the primary cellular function of AdipoR1 and AdipoR2 is to maintain membrane fluidity (Ruiz M et al., J Lipid Res. 2019 May;60(5):995-1004). Loss of them might disturb membrane fluidity and thereby decrease signaling. Another explanation may be intracellular signaling by HMW-adiponectin internalization through binding to T-cadherin without interaction with AdipoRs. In our future studies, we will carefully examine whether HMW-adiponectin stimulates such signaling or not and whether endocytosis of adiponectin through T-cadherin is required or not.

Reviewer #3:I was ambivalent about this manuscript the first time around and the authors have only improved it with the additional experiments. In all honesty, this will never be the cleanest study and will not have the highest level of impact due to lack of mechanism, whatever the authors do to address the issues. However, I continue to believe that it brings up a valid point about the ambiguity of the true adiponectin receptor and the data are strong enough to provide a valid challenge to the dogma. On balance, I recommend acceptance.

Thank you very much for evaluating the significance of our study. Regarding the mechanism, how adiponectin input intracellular signaling, it must be the next important study. It is well known that adiponectin stimulates intracellular signaling cascades such as phospho-AMPK and phospho-p38. And the loss of AdipoRs decreases these signaling. One explanation will be the generation of low molecular weight form of adiponectin such as globular form after binding to T-cadherin. However, it was recently reported that the primary cellular function of AdipoR1 and AdipoR2 is to maintain membrane fluidity (Ruiz M et al., J Lipid Res. 2019 May;60(5):995-1004). Loss of them might disturb membrane fluidity and thereby decrease signaling. Another explanation may be intracellular signaling by HMW-adiponectin internalization through binding to T-cadherin without interaction with AdipoRs. In our future studies, we will carefully examine whether HMW-adiponectin stimulates such signaling or not and whether endocytosis of adiponectin through T-cadherin is required or not.